# Vascular Endothelial Growth Factor as Molecular Target for Bronchopulmonary Dysplasia Prevention in Very Low Birth Weight Infants

**DOI:** 10.3390/ijms24032729

**Published:** 2023-02-01

**Authors:** Serafina Perrone, Sara Manti, Luca Buttarelli, Chiara Petrolini, Giovanni Boscarino, Laura Filonzi, Eloisa Gitto, Susanna Maria Roberta Esposito, Francesco Nonnis Marzano

**Affiliations:** 1Neonatology Unit, Pietro Barilla Children’s Hospital, Department of Medicine and Surgery, University of Parma, Via Gramsci 14, 43126 Parma, Italy; 2Department of Human Pathology in Adult and Developmental Age “Gaetano Barresi”, Unirsity of Messina, Via Consolare Valeria 1, 98125 Messina, Italy; 3Pediatric Clinic, Pietro Barilla Children’s Hospital, Department of Medicine and Surgery, University of Parma, Via Gramsci 14, 43126 Parma, Italy; 4Department of Chemistry, Life Sciences and Environmental Sustainability, University of Parma, Viale delle Scienze 11, 43125 Parma, Italy

**Keywords:** BPD, oxidative stress, preterm, VEGFA, biomarker, inflammation, lung disease

## Abstract

Bronchopulmonary dysplasia (BPD) still represents an important burden of neonatal care. The definition of the disease is currently undergoing several revisions, and, to date, BPD is actually defined by its treatment rather than diagnostic or clinic criteria. BPD is associated with many prenatal and postnatal risk factors, such as maternal smoking, chorioamnionitis, intrauterine growth restriction (IUGR), patent ductus arteriosus (PDA), parenteral nutrition, sepsis, and mechanical ventilation. Various experimental models have shown how these factors cause distorted alveolar and vascular growth, as well as alterations in the composition and differentiation of the mesenchymal cells of a newborn’s lungs, demonstrating a multifactorial pathogenesis of the disease. In addition, inflammation and oxidative stress are the common denominators of the mechanisms that contribute to BPD development. Vascular endothelial growth factor-A (VEGFA) constitutes the most prominent and best studied candidate for vascular development. Animal models have confirmed the important regulatory roles of epithelial-expressed VEGF in lung development and function. This educational review aims to discuss the inflammatory pathways in BPD onset for preterm newborns, focusing on the role of VEGFA and providing a summary of current and emerging evidence.

## 1. Introduction

Despite more than five decades having passed, bronchopulmonary dysplasia (BPD) still represents an important burden of neonatal care. The definition of the disease is undergoing several revisions, and for this reason, BPD is actually defined by its treatment rather than diagnostic or clinic criteria [1]. According to the National Institute of Health [1], BPD is a chronic lung disease affecting infants born before 32 weeks of gestational age (GA), with radiological evidence of damage to the lung parenchyma, and requiring respiratory support (either invasive or non-invasive) at 36 weeks post-menstrual age for at least three or more consecutive days to maintain a peripheral arterial oxygen saturation (SpO_2_) >90% [2,3,4].

BPD is associated with many prenatal risk factors, including maternal smoking, chorioamnionitis, and intrauterine growth restriction (IUGR), in addition to postnatal risk factors, such as patent ductus arteriosus (PDA), parenteral nutrition, sepsis, and mechanical ventilation [5,6,7,8,9,10]. Respiratory models of premature infants can also be considered risk factors for the development of BPD. In this regard, one study has suggested that ventilatory peak inspiratory pressure and the need for assisted ventilation on the 4th day of postnatal life can be considered early indicators of BPD [11]. Other subsequent studies have also shown that, during the first two weeks of life in extremely preterm infants, three models of chronic lung disease can emerge: one model is characterized by a low inspiratory fraction of oxygen (FiO_2_), one model is characterized by pulmonary deterioration (PD) following initial improvement in the first few days, and the last one is characterized by early persistent pulmonary deterioration requiring prolonged respiratory support from birth [12]. Accordingly, in a previous study, after measuring FiO_2_ during the first seven days of life and then on the fourteenth day of life, it was noted that almost 50% of children in the second group and almost 67% of the third group developed BPD [13].

BPD is a common health complication among extremely preterm infants [14], and its wide epidemiological range (10–89%) well reflects the differences in the GA and birth weight of newborns across study populations, as well as the differences in diagnostic criteria and care practices across study institutions and localities. Therefore, it is reasonable to affirm that the incidence of BPD is inversely related to GA at birth, with higher incidences reported with decreasing GA [15]. Furthermore, small-for-GA (SGA) infants have a higher incidence of BPD than adequate-for-GA (AGA) preterm infants, especially between 26 and 28 weeks of PMA at birth. As SGA infants are overrepresented among preterm births, there is likely a multiplier effect on BPD incidence that must be accounted for. BPD can also be seen as one of the main causes that lead extremely low birth weight infants (ELBW) to have a higher mortality and morbidity than very low birth weight (VLBW) infants [2,14,16].

In addition, inflammation is a common denominator of many factors that contribute to BPD development, and it seems to be an important risk factor for the pathogenesis of the disease [2]. In fact, the overexpression of pro-inflammatory cytokines is counterbalanced by the suppression of growth-promoting cytokines. Vascular endothelial growth factor A (VEGFA) constitutes the most prominent and best studied candidate for vascular development [2,17] (Figure 1).

In view of the above cited studies, in this work, we aimed to analyze, in an educational review, the inflammatory pathways in the genesis of BPD for preterm newborns, focusing on the role of VEGFA.

## 2. Lung Development

Proper lung development and function are essential for neonatal transition from intrauterine to extrauterine life. The lungs have the highest levels of vasculature and vascular endothelial growth factor (VEGF) expression [18]. An adequate amount of VEGF expression is essential for pulmonary development.

VEGF contributes to the formation of vascular beds during embryo development, as demonstrated by the lethality of single-allele VEGF knockout due to abnormal vasculogenesis [19].

Many different lung cells produce VEGF and also respond to VEGF [20]. In the lungs, VEGF functions as a mitogen and as a survival and differentiation factor for endothelial cells [21]. For example, type II pneumocytes undergo growth and differentiation in the presence of VEGF [22,23].

A decreased level of VEGF may contribute to BPD and to infants dying with abnormal alveolar microvessels [24].

BPD is a disease of preterm newborns, a vulnerable population with blocked lungs and who are in a phase of immature development: the bronchioles, precapillaries, and mucous glands are not yet fully developed, the blood–air barrier is still not properly formed, and surfactant production has not yet begun [2,25].

Various experimental models have shown how infections, oxygen, and mechanical ventilation cause distorted alveolar and vascular growth, as well as alterations in the composition and differentiation of the mesenchymal cells of a newborn’s lungs [26,27]. Assisted ventilation, required to begin breathing in preterm lungs, can damage the lungs via exposure to high pressures, volumes, and oxygen. These factors can activate inflammatory pathways that can amplify any pre-existing injuries or perpetuate the inflammatory mechanisms triggered by the same ventilator support therapy [2]. The combination of a hyperoxic picture from oxygen administration with hypoxemic events, frequent in preterm babies, increases the production of reactive oxygen species (ROS) and consequent tissue inflammation [2,27].

### 2.1. Lung Inflammation

Inflammation is a common denominator of many factors that contribute to BPD development.

The inflammatory condition concerning BPD is the result of an imbalance between pro-inflammatory and anti-inflammatory mechanisms, with a persistent imbalance toward the former [2]. The pro-inflammatory cytokines involved in the pathogenesis of BPD are mainly interleukin (IL)-1β, IL-6, IL-8, and tumor necrosis factor-α (TNF-α). More precisely, altered levels of pro-inflammatory cytokines (i.e., IL-6, TNF-α, IL-1β, IL-8, and IL-10) have been reported in different samples of the amniotic fluid [28], cord blood [29], and tracheal aspirate of newborns with a BPD diagnosis [30]. Conversely, lower levels of anti-inflammatory cytokines, such as IL-10, are reported [2].

Among the pro-inflammatory cytokines, IL-1β has been proven to be associated with the subsequent development of BPD in premature infants. Its role seems to be linked to the modification of the expressions of the proteins involved in mediating the cellular response to retinoic acid [31,32]. The concentration of IL-8 has been found to be much higher in the homogenates of pre-term explant cultures of lungs exposed to hyperoxia than in those of the lungs of full-term babies [33]. High TNF-α levels promote chronic inflammation, leading to BPD, as shown in the study by Oncel et al., where the inhibition of TNF-α was found to be beneficial for alveolar and lung injury by reducing lung inflammation and oxidative stress [34]. However, the deficiency of an anti-inflammatory cytokine such as IL-10 is associated with BPD severity according to Mao et al. [35].

Interestingly, studies have shown an early increase in pro-inflammatory cytokines levels during the stages of lung development, as assessed by detecting pro-inflammatory cytokines in the tracheal aspirates of patients with BPD, whereas levels of IL-10 in both the serum and tracheal aspirates have been shown to be decreased in infants who developed BPD [36]. These pro-inflammatory cytokines promote the formation of pulmonary edema.

In preterm infants who died of severe respiratory distress syndrome (RDS), the influx of TNF-α-positive macrophages in pulmonary tissue was found to be associated with a striking loss of endothelial basement membranes and a destruction of interstitial glycosaminoglycans [37]. It has been demonstrated that these mediators attract cells of innate immunity: preterm infants at various BPD stages were found to have a much higher and more persisting number of neutrophils and macrophages in their bronchoalveolar lavage fluid than infants who had recovered from RDS [38]. Neutrophils exert substantial lung damage through the release of proteases, such as metalloproteases and elastases. 

The inflammatory state in patients with BPD also results in an increased production of beta tissue growth factor (TGF-β). Therefore, both the chronic inflammatory picture and the strongly oxidative environment due to respiratory support are the main factors in the pathogenesis of BPD. However, in order for BPD to occur, environmental factors, such as chorioamnionitis, neonatal sepsis, and hyperoxia, are also required to promote and sustain a persistent inflammatory status [2,27,36,38]. In addition, the airway microbiome and metabolome could influence the genesis and support of BPD [39]. A decreased diversity of the airway microbiome, specifically with an abundance of *Stenotrophomonas*, and an increased level of sn-glycerol 3-phosphoethanolamine were recently reported in patients with BPD [39]. The authors found a relationship between the occurrence and severity of the disease and *Stenotrophomonas,* concluding that this bacterium may influence the composition of the lower airway microbiome through its metabolite sn-glycerol 3-phosphoethanolamine and that it may be the triggering factor for the disease, suggesting the use of this factor as a novel biomarker of BPD.

Studies have shown that a preterm neonate with structurally and functionally immature lungs (at the canalicular/saccular phase of lung development), paired with an underdeveloped immune response, exhibits an increased pro-inflammatory response when BPD develops [38]. Therefore, the precocious modulation of inflammatory processes might turn out to be a valid preventive and/or therapeutic approach for BPD [35,40].

### 2.2. Hyperoxia and Oxidative Stress

Oxidative stress in the lungs occurs when the antioxidant capacity is overwhelmed or depleted through external exposures, such as altered oxygen tension or air pollution, or internally. The internal sources of oxidative stress include systemic disease and the activation of resident and inflammatory cells recruited in response to an exposure or systemic reaction [41]. This situation can occur more easily in preterm infants since their antioxidant systems are still immature, and the so-called free-radical-related diseases (FRDs), such as BPD, can easily occur [42].

Although mechanical ventilation and, therefore, the use of oxygen support are the main therapeutic strategies in managing BPD, they are also the main risk factors for BPD [27]. Unlike at-term infants, preterm ones (and especially those with VLBW) have an imperfect anti-oxidative stress system due to pulmonary insufficiency, a lack of alveolar surfactant, and endogenous antioxidant enzyme system defects; all these events make babies more likely to be exposed to hyperoxia. The high oxygen exposure triggers a cascade of oxidative stress and induces an influx of inflammatory cells. This kind of damage may develop to ventilator-induced lung injury via volutrauma from the administration of large volumes of gas to the lungs; barotrauma from high airway pressures; atelectrauma due to repeated alveolar collapse and re-expansion; and the release of inflammatory mediators of the damaged alveolar epithelium, such as IL-33, whose levels have been found to increase when exposed to mechanical ventilation with oxygen-rich air (MV-O_2_) [43,44]. Hyperoxia-exposure-mediated lung injury mainly includes cell necrosis or apoptosis in the alveolar epithelium and vascular endothelium, the destruction of the alveolar structure, increased vascular permeability, and the reduced formation of capillaries in the distal lung tissue, as well as the massive recruitment of inflammatory cells, which, in turn, may delay the separation of cystic alveoli, inducing the proliferation of stromal cells and, thus, resulting in the persistence of inflammation. Lastly, all these events could result in pulmonary hypertension [36,41,43,44,45].

The above-described cell damage is induced via reactive oxygen species (ROS) production by neutrophils. Specifically, the recruited neutrophils release ROS, which, in turn, can cause tissue damage by modulating lipid peroxidation and inducing antiprotease inhibitors [46] (Figure 2).

The biomarkers of oxidative stress include increased myeloperoxidase activity and xanthine oxidase quantities, which derive from phagocyte activation and hypoxia–reoxygenation mechanisms, respectively (Figure 3).

The concentrations of these markers are increased during the first week of life in the tracheal aspirates of infants who develop BPD compared with the concentrations in infants who are ventilated but recover without BPD [47].

Therefore, studies have included antioxidants as one of the potential therapies for BPD: many agents have been shown to be protective in animal models, but only few substances have been used for newborns in pilot studies. The aim of this kind of therapy may be to develop antioxidant medication that not only protects against the direct injurious effects of oxidants but that also fundamentally alters the associated inflammatory events [42,48].

Oxidative stress also plays an important role in angiogenesis by influencing the activities of VEGF. Some studies have demonstrated that exogenous ROS stimulate the induction of VEGF expression in various cell types, such as endothelial cells, smooth muscle cells, and macrophages, whereas VEGF induces endothelial cell migration and proliferation through an increase in intracellular ROS. In some oxygen-sensitive tissues, such as the retina, ROS can trigger a continuous production of VEGF, which can be of central importance for the onset and development of retinal conditions. Oxygen species also affect VEGF-stimulated VEGFR2 dimerization and autophosphorylation that are required for VEGFR2 activation and subsequent angiogenesis [49,50].

### 2.3. Vascuologenesis

The underlying mechanisms driving pulmonary vascular growth in BPD still remain under debate. In the early stages of lung development, vessel growth follows two pathways: angiogenesis, characterized by the extension of existing vessels, and vasculogenesis, characterized by the differentiation of angioblasts and hemangioblasts into de novo vascular structures [51,52]. During the branching morphogenesis of the airways, pulmonary vascular structures form in close proximity, suggesting that the airways may form a template for early vascular development. Later, the further growth of the vascular networks allows for blood flow, in parallel with the division of the alveoli into complex acinar, throughout infancy and adolescence [53,54].

In patients with BPD, the cessation of growth of preterm lungs leads to a decreased capillary density, a smaller cross-sectional area for blood circulation, and, lastly, a reduced surface area for gas exchange. In addition, the presence of fully developed vessels with dysmorphic anastomoses has also been reported in the lungs of babies with severe BPD. Intrapulmonary arteriovenous anastomotic vessels prevent gas exchange at the alveolar–capillary interface, resulting in persistent hypoxemia, the vasoconstriction of the arteries, and the further progression of tissue damage [55,56].

To counteract the “dysmorphic” pattern characterized by the paucity of capillaries within abnormally enlarged alveoli, abnormal collateral circulations appear but worsen the lung injury. Moreover, abnormal muscularization and a heightened vascular tone have also been reported to contribute to BPD onset [57,58]. 

Moreover, changes in circulating and resident endothelial progenitor cell (EPC) levels have been associated with BPD onset [59,60]. Similarly, increased mesenchymal stromal cell expressions in tracheal aspirates were found to be associated with the development of severe BPD [61,62]. The authors demonstrated that MSCs are able to produce TGF-β1 in an autocrine manner, which, in turn, promotes the differentiation of MSCs into myofibroblasts with subsequent fibrotic tissue degeneration [63,64]. 

Interestingly, it seems that these events occur more frequently in genetically predisposed subjects, although genome-wide association studies have shown controversial findings [65,66].

## 3. Vascular Endothelial Growth Factor

VEGF is a highly specific mitogen for vascular endothelial cells with additional effects on several cell types. Various VEGF isoforms are generated as a result of the alternative splicing of a single VEGF gene that is located on chromosome 6p21.3 and consists of eight exons exhibiting alternating splicing and constituting a protein family. As shown by an analysis of the 5′ region, the human gene for VEGF is highly polymorphic, and, for this reason, it seems likely to be a contributor to the hereditary predisposition of diseases in which angiogenesis plays a role. Among the various polymorphisms, the genotype +936 C/T is the most studied for various pathologies [51,52,53,54,55,56,57].

The importance of VEGF as a central regulator of vasculogenesis has been demonstrated in studies using the targeted gene disruption technique in mice [59]. Animals without one of the two VEGF alleles die before birth due to defects in the development of the cardiovascular system. Similarly, the disruption of genes encoding the receptor tyrosine kinases (RTKs) of VEGF, VEGFR1, and VEGFR2 causes serious abnormalities in the formation of blood vessels in homozygous animals [59]. Embryos without the VEGFR2 gene die before birth because endothelial cell differentiation does not occur, and blood vessels do not form. The main stimuli that produce VEGF are hyperglycemia and hypoxia through the link with hypoxia-induced factor 1 (HIF-1), but factors that are released in inflammatory contexts (IL-1β, TGF β, and TNF-α) can also lead to an increase in VEGF expression. The expressions of VEGFR1 and VEGFR2 receptors are also regulated by hypoxia [51,59,61].

Regarding the fetal development in which VEGF plays a key role, the lung is the site with the highest levels of vascularization and VEGF expression. The large pulmonary capillary bed is essential for an efficient exchange of gas between the alveoli and the capillaries. However, the vascular side is not the only one affected by the properties of VEGF: the transient inactivation of the VEGF lung gene leads to the apoptosis of cells in the alveolar septum wall, the enlargement of the airspace, and an increase in lung compliance. In addition, type II cells, responsible for surfactant production, go through growth and differentiation processes in the presence of VEGF. Previous animal model studies have shown that a transient decrease in pulmonary VEGF leads to increased alveolar and bronchial cell apoptosis, air space enlargement, and changes in lung elastic recoil (processes that are characteristic of emphysema) that persist for at least 8 weeks [63,65].

Lassus et al. demonstrated that the increasing pulmonary concentrations of VEGF in preterm infants suggested a physiologic role for this growth factor in preterm lungs [67]. The authors observed a high VEGF expression in tracheal fluid, and a further increase was detected every postnatal day, while plasmatic levels remained constant. However, patients developing BPD showed lower VEGF levels during this period; this was not due to their lower gestational age because any significant correlation between VEGF in tracheal aspirate samples and GA was reported. The infants who eventually developed BPD required higher inspiratory oxygen concentrations [67]. In another study, hyperoxia was reported to decrease VEGF expression in alveolar epithelial cells in rabbits [17]. This study agrees with these previous observations since lower VEGF concentrations were found in infants who later developed BPD [65,67]. These findings are in line with the data reported by Mariduena et al. [68]. In their study, the authors showed that low serum VEGF levels were detected in patients with BPD during the first week of postnatal life [68].

Bhatt et al. [24] compared premature infants who died with BPD with term infants who died of non-pulmonary causes and who had a short exposure to supplemental oxygen and mechanical ventilation. Immunohistochemistry testing for VEGF expression in these infants found a decrease in the VEGF protein in BPD lungs, particularly in areas with thickened alveolar septa. The cell type expressing the VEGF mRNA had not been modified. These data support the hypothesis that a decrease in VEGF may play a role in stopping the development of pulmonary microcirculation in BPD since infants dying with BPD have abnormal alveolar microvessels that are consistent with disrupted alveolar vascular development [24]. Previous studies have shown that in vitro and in vivo type II pneumocytes that express high levels of surfactant protein C (SP-C) have low levels of VEGF. Bhatt et al. showed that BPD lungs had more type II cells expressing SP-C [69,70]. Type II pneumocytes expressing VEGF may have been replaced by cells expressing SP-C as part of the damage repair process [24]. However, further studies are advocated to confirm this thesis. 

A reduction in VEGFR2 receptor activity can also cause serious damage to lung development. In an experimental model, Le Cras et al. [70] reported that a lower VEGF receptor activity is involved in the early onset of pulmonary hypertension. It is therefore hypothesized that the early interruption of the signaling of VEGFR in a newborn’s lungs might contribute to the pathological sequelae of BPD and to a long-term predisposition for adult diseases, such as chronic obstructive pulmonary disease (COPD) and pulmonary hypertension [71]. Accordingly, an experimental model reported that the transient interruption of VEGF signaling decreased the bioavailability of nitric oxide (NO) in the lungs of newborn rats, which may contribute to reduce lung growth and pulmonary hypertension [72]. The neonatal inhibition of the VEGF receptor immediately and transiently induced apoptosis in pulmonary vascular endothelial cells, which was followed by the persistent impairment of vascular growth and alveolysis in newborn rats. In addition, early treatment with inhaled NO after the inhibition of VEGF reduced the apoptosis of endothelial cells in neonatal rat lungs [71,72]. However, based on the results of recent meta-analyses and systematic reviews, there has been no long-term improvement in the mortality or the incidence and severity of BPD with the use of iNO in preterm infants as prevention or therapy [73].

Nowadays, genetic factors are becoming one of the main focuses of interest [74]. Many studies have been conducted regarding a genetic component as a possible cause for BPD and, even though they have not found a connection with their respective “genetic targets”, they have pointed to various variants, genes, or pathways associated with the susceptibility of BPD, making it likely that each genetic variant plays a minimal role in BPD development [72].

However, in 181 newborns with a mean gestational age of 28 wks, Kwinta et al. [75] discovered that a particular polymorphism of VEGF (VEGF-460T>C) may influence the risk of BPD.

Conversely, Filonzi et al. [76] analyzed the possible relationship between the +936 C/T VEGF polymorphisms and the -710 VEGFR1 C/T polymorphisms and the frequency of BPD among eighty-two very low birth weight infants, without major anomalies. In their study, 33 out of 82 infants developed BPD (the BPD group), and 49 infants without BPD served as controls (the control group). Although significant statistic differences were found between the newborns with BPD and the controls with regard to gestational age, birth weight, mechanical ventilation, the duration of oxygen therapy, maternal preeclampsia, and chorioamnionitis, no differences were detected between genotypic and allelic levels regarding the VEGFR1 and VEGF molecular polymorphisms. Thus, no association between those genes and BPD was found. This supports the hypothesis that BPD is a heterogeneous disease resulting from multiple gene interactions and pathways. In a retrospective study, Poggi et al. [77] investigated the roles of specific polymorphisms of genes encoding for VEGF-A in a cohort of preterm infants, and they correlated their presence with the development of BPD [77]. The authors reported that VEGF-A might independently affect birth weight and gestational age and act as a protecting or risk marker for prematurity complications, such as BPD [77].

Additionally, other possible causes of BPD, such as pulmonary infection and nutrition status and their correlation with VEGF, have been hypothesized (Figure 4).

In an animal model of intrauterine infection/inflammation with pregnant rats endocervically inoculated with *Escherichia coli*, Pan et al. [78] reported that an impaired pseudoglandular stage was associated with a significantly decreased VEGF expression in the lung tissue of fetal rats and neonatal rats, confirming the crucial role of VEGF in lung development.

An extremely low birth weight and a reduced caloric intake contribute to BPD development. In this regard, suboptimal nutrition negatively affects lung VEGF expression in formula-fed neonatal rats subjected to hyperoxia [79]. However, when supplemented with probiotics, the formula-fed neonatal rats showed an unchanged VEGF expression with the preservation of angiogenesis [79].

Similarly, omega-3 polyunsaturated fatty acid (PUFA ω-3) supplementation prevents BPD-associated pulmonary hypertension by reversing the reduced VEGFA and VEGF levels [80].

## 4. Conclusions

BPD is a condition whose pathogenesis involves a large number of factors and pathways, and VEGF plays an important role in both lung development and function. Released principally by respiratory epithelial cells, VEGF enhances the migration, proliferation, and differentiation of endothelial cells via paracrine signaling, thus appearing to be essential for maintaining air–blood structures. The recognition of the central role of VEGF in angiogenesis and its interplay with oxidative stress has led to the hypothesis that its modulation may represent a potential treatment of conditions characterized by pathologic angiogenesis. Moreover, having the possibility to modulate the levels of VEGF, for example, by diet supplementation, could represent a new therapeutic strategy to improve endothelial dysfunction, especially in newborns at a high risk of BPD. Furthermore, investigating VEGF’s role and function could allow for determining the link with lipid metabolism, whose exact mechanisms and role in BPD are still relatively obscure.

In conclusion, the new evidence suggests that VEGF levels represent a promising therapeutic approach for preventing neonatal bronchopulmonary dysplasia and its adverse outcomes [2,81]. Further clinical studies are needed to evaluate the molecular target for drug research.

## Figures and Tables

**Figure 1 ijms-24-02729-f001:**
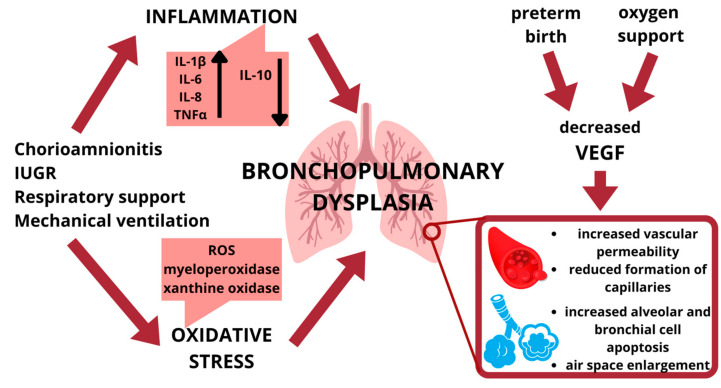
Schematic representation of risk factors associated with bronchopulmonary dysplasia.

**Figure 2 ijms-24-02729-f002:**
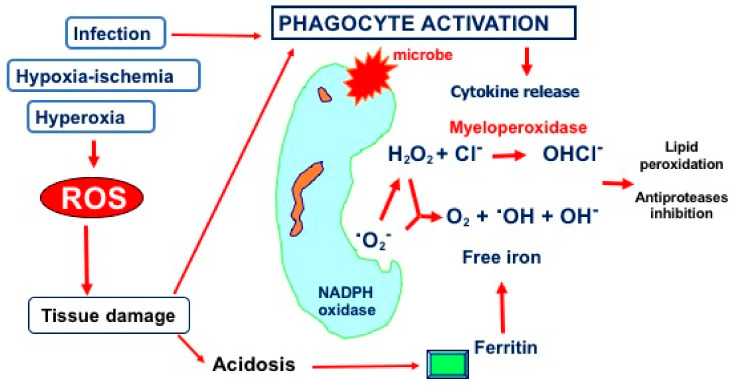
Schematic representation of free radical generation during phagocyte activation. Following injury, due to infection, hypoxia–ischemia, or hyperoxia, neutrophils release ROS. The superoxide anion (^·^O_2_^−^), the most abundant radical species, is the first stage of bacterial killing reaction, which is followed by generation of other ROS, such as hydroxyl radical (^·^OH) by free irons and hypochlorous acid (OHCL^−^) by myeloperoxidases.

**Figure 3 ijms-24-02729-f003:**
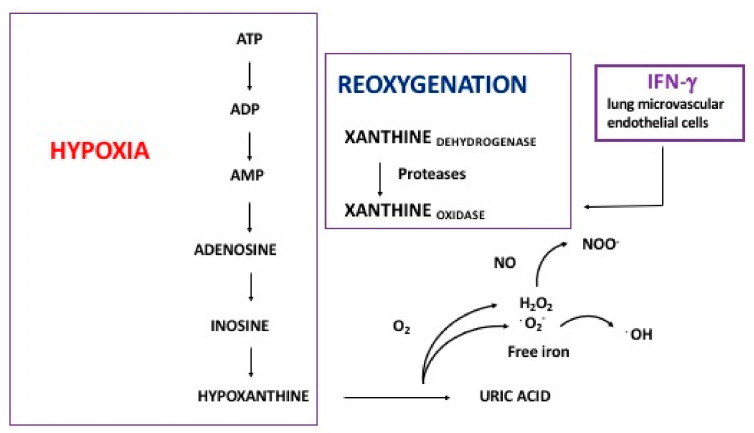
Schematic representation of free radical generation during hypoxia–reoxygenation. Hypoxanthine derives from degradation of adenosine 5′-triphosphate (ATP) during hypoxia-induced anaerobic metabolism. During reoxygenation, xanthine oxidoreductase catalyzes hydroxylation of hypoxanthine to xanthine and uric acid, inducing the release of ROS. Xanthine oxidoreductase exists in two forms: xanthine dehydrogenase and xanthine oxidase. An irreversible proteolytic conversion of xanthine dehydrogenase to xanthine oxidase can also be specifically induced by IFN-γ in lung microvascular endothelial cells.

**Figure 4 ijms-24-02729-f004:**
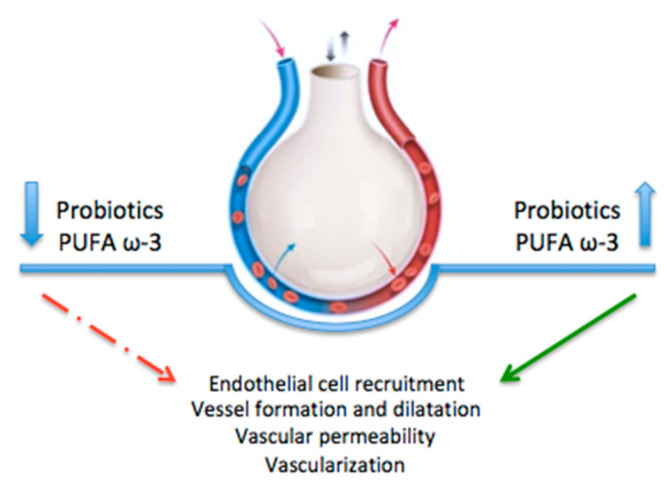
Diet supplementation and vascular development. Supplementation with probiotics and PUFA ω-3 affect endothelial cell recruitment, vessel formation and dilatation, vascular permeability, and vascularization.

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
