# Peer review of "Vascular Endothelial Growth Factor as Molecular Target for Bronchopulmonary Dysplasia Prevention in Very Low Birth Weight Infants"

_ijms, 2023, doi:10.3390/ijms24032729_

Round 1

Reviewer 1 Report

The review article entitled “Vascular endothelial growth factors as molecular target for bronchopulmonary dysplasia prevention in very low birth weight infants” has been reviewed. The authors have made a very clear description about the pathophysiology of bronchopulmonary dysplasia and the recent molecular finding about bronchopulmonary dysplasia. The authors mentioned 3 possible mechanisms about lung development and relationship with bronchopulmonary dysplasia: lung inflammation, hyperoxia and oxidative stress, vascuologenesis. These mechanisms were all related to VEGF deficiency. However, there were some other possible causes of bronchopulmonary dysplasia, such as pulmonary infection and nutrition status. If some more discussion about these factors and the possible correlation with VEGF, it would be much better.

Author Response

We thank the reviewer for the comments.

Following the suggestion, the relationship bewteen BPD-pulmonary infection and VEGF and on  BPD-nutrition status  and VEGF has been added in the text 

In addition a new Figure (Figure 4 in the manuscript) has been provided 

Reviewer 2 Report

Line 195: “The above-described cell damage is induced by reactive oxygen species (ROS) production by neutrophils.” Figure 1 mentions “myeloperoxidase” and “xanthine oxidase” and Line 199 states “Biomarkers of oxidative stress include increased myeloperoxidase activity and quantities of xanthine oxidase”. It is unclear from these statements whether ROS are generated by neutrophils or by myeloperoxidase/xanthine oxidase.

Line 196: “ROS cause tissue damage by lipid peroxidation, potentiate tissue damage by inhibiting antiproteases that modulate the activity of proteases such as elastase.” Is this a complete sentence? Do the authors want to state that ROS inhibit antiproteases? Please clarify this sentence.

Line 347: “4. Conclusions” seem to be that “further studies are advocated ….”. Can the conclusion be more definitive in the role of VEGF as a molecular target for BPD based on scientific evidence this article describes?

More figures should be added to this paper.

Author Response

Line 195: “The above-described cell damage is induced by reactive oxygen species (ROS) production by neutrophils.” Figure 1 mentions “myeloperoxidase” and “xanthine oxidase” and Line 199 states “Biomarkers of oxidative stress include increased myeloperoxidase activity and quantities of xanthine oxidase”. It is unclear from these statements whether ROS are generated by neutrophils or by myeloperoxidase/xanthine oxidase.

Re: we thank the reviewer for this remark. The mechanisms of production of reactive oxygen species have been better clarified in the manuscript. 

Line 196: “ROS cause tissue damage by lipid peroxidation, potentiate tissue damage by inhibiting antiproteases that modulate the activity of proteases such as elastase.” Is this a complete sentence? Do the authors want to state that ROS inhibit antiproteases? Please clarify this sentence.

Re: we thank the reviewer for this remark. The sentence has been rewritten and two figures have been added. 

Line 347: “4. Conclusions” seem to be that “further studies are advocated ….”. Can the conclusion be more definitive in the role of VEGF as a molecular target for BPD based on scientific evidence this article describes?

Re: many thanks for these observations. The conclusion paragraph has been rewritten. 

More figures should be added to this paper.

Re: three figures have been added in the text

Round 2

Reviewer 2 Report

Line 215: ".... by modulating the lipid peroxidation and inducing the inhibitors of antiproteases [Figure 2]. -- Figure 2 does not indicate "lipid peroxidation" or "antiproteases".   Line 222: "hydroxic radical" should be "hydroxyl radical". "(OH-)" should be "(.OH)". "(OHCL)" should be "(OHCl-).   Figure 3: Should "XANTINA REDUCTASE" be "XANTHINE DEHYDROGENASE"? Should "XANTINA OXIDASE" be "XANTHINE OXIDASE"?   Figure 3: Should "O." be ".OH"?

Author Response

Line 215: ".... by modulating the lipid peroxidation and inducing the inhibitors of antiproteases [Figure 2]. -- Figure 2 does not indicate "lipid peroxidation" or "antiproteases".   Line 222: "hydroxic radical" should be "hydroxyl radical". "(OH-)" should be "(.OH)". "(OHCL)" should be "(OHCl-).   Figure 3: Should "XANTINA REDUCTASE" be "XANTHINE DEHYDROGENASE"? Should "XANTINA OXIDASE" be "XANTHINE OXIDASE"?   Figure 3: Should "O." be ".OH"?

RE: we thank the reviewer for these important remarks. The mistakes have been corrected in the text. Figure 2 and 3 have been modified. 

Round 3

Reviewer 2 Report

.